# Combining Linkage and Association Mapping Approaches to Study the Genetic Architecture of Verticillium Wilt Resistance in Sunflower

**DOI:** 10.3390/plants14081187

**Published:** 2025-04-11

**Authors:** Juan F. Montecchia, Mónica I. Fass, Matías Domínguez, Sergio A. González, Martín N. García, Carla V. Filippi, Emiliano Ben Guerrero, Carla Maringolo, Carolina Troglia, Facundo J. Quiroz, Julio H. González, Daniel Alvarez, Ruth A. Heinz, Verónica V. Lia, Norma B. Paniego

**Affiliations:** 1Instituto de Agrobiotecnología y Biología Molecular—IABIMO—INTA-CONICET, Instituto de Biotecnología, Centro de Investigaciones de Ciencias Veterinarias y Agronómicas, INTA, Hurlingham B1686, Argentina; fass.monica@inta.gob.ar (M.I.F.); gonzalez.sergio@inta.gob.ar (S.A.G.); garcia.martin@inta.gob.ar (M.N.G.); carlavfilippi@gmail.com (C.V.F.); benguerrero.emiliano@inta.gob.ar (E.B.G.); heinz.ruth@inta.gob.ar (R.A.H.); 2Advanta Semillas S.A.I.C., Estación Experimental Venado Tuerto, Ruta Nac. 33 km 636, Venado Tuerto PC 2600, Argentina; 3Estación Experimental Agropecuaria Pergamino, INTA, Av. Frondizi km 4.5, Pergamino B2700, Argentina; dominguez.matias@inta.gob.ar (M.D.); julionaveiro@gmail.com (J.H.G.); 4Laboratorio de Bioquímica, Departamento de Biología Vegetal, Facultad de Agronomía, Universidad de la República, Avenida Garzón 780, Montevideo 12900, Uruguay; 5Estación Experimental Agropecuaria Balcarce, INTA, Ruta 226 km 73.5, Balcarce B7620, Argentina; maringolo.carla@inta.gob.ar (C.M.); troglia.carolina@inta.gob.ar (C.T.); quiroz.facundo@inta.gob.ar (F.J.Q.); 6Estación Experimental Agropecuaria Manfredi, INTA, Ruta Nac. 9 km 636, Manfredi X5988, Argentina; alvarez.daniel@inta.gob.ar; 7Facultad de Ciencias Exactas y Naturales, Universidad de Buenos Aires, Intendente Güiraldes 2160, Ciudad Autónoma de Buenos Aires C1428, Argentina

**Keywords:** *Verticillium dahliae*, biparental QTL mapping, genome-wide association studies, ddRADseq, crop disease resistance

## Abstract

Sunflower Verticillium Wilt and Leaf Mottle (SVW), caused by *Verticillium dahliae* Kleb., is a globally prevalent disease affecting sunflower production. In this study, we identified a major quantitative trait locus (QTL) on chromosome 10 and other genomic regions associated with SVW resistance by integrating biparental and association mapping in sunflower populations from the National Institute of Agricultural Technology. Nine replicated field trials were conducted in highly infested *V. dahliae* reservoirs to assess disease incidence and severity. Both mapping populations were genotyped using double-digest restriction-site-associated DNA sequencing (ddRADseq). Association mapping with 18,161 SNPs and biparental QTL mapping with 1769 SNPs identified a major QTL on chromosome 10 explaining up to 30% of phenotypic variation for disease incidence at flowering and for the area under the disease progress curve for disease incidence, and which contributes to a lesser extent to disease severity reduction. Additional QTLs on chromosomes 17, 8, 9, 14, 13, and 11 were associated with reduced disease incidence, severity, or both. Candidate genes were identified within these associated regions, 39 of which are in the major QTL on Chromosome 10. These findings demonstrate the value of integrating complementary QTL mapping strategies for validating resistance loci and advancing sunflower breeding for SVW resistance.

## 1. Introduction

Sunflower Verticillium Wilt and Leaf Mottle (SVW) is a monocyclic vascular disease caused by the soil-borne fungal pathogen *Verticillium dahliae* (Kleb.). It occurs in most sunflower-producing regions worldwide and has significantly impacted large areas in Argentina, Canada, and the United States [1]. In recent years, SVW has emerged as a serious threat to sunflower production in China [2] and temperate European countries, with increasing prevalence reported in France, Italy, Spain, and the Black Sea region [3,4].

SVW-induced wilting has been shown to reduce yields by up to 30% in susceptible commercial sunflower hybrids [5,6] and by as much as 73% in highly susceptible materials grown in severely infested fields [7]. Yield losses are directly correlated with symptom severity, particularly the extent of foliage necrosis [6,8].

SVW resistance was initially described as a qualitative trait controlled by a single dominant locus, V1, which became the primary source of resistance used in hybrid development worldwide [9,10]. Subsequent studies have identified various inbred lines (ILs) carrying dominant, additive, and recessive resistance sources against North American *V. dahliae* races [11,12]. However, *V. dahliae* races capable of overcoming V1-conferred resistance were first reported in Argentina [13,14] and later in the United States and Europe [15,16].

Argentina, the world’s third-largest producer and exporter of sunflower edible oil [17,18], harbors *V. dahliae* as an endemic pathogen with diverse local races [13,14,19]. The pathogen’s inoculum is distributed across 1.2 million hectares, affecting over 70% of the country’s sunflower-growing regions [20]. In the southern region of Buenos Aires Province, Argentina’s primary sunflower-producing area, SVW prevalence has averaged 45% (±14%) over the last decade [20]. Two major local *V. dahliae* races, VArg1 and VArg2, have been identified, prompting the development of differential inbred lines with race-specific resistance and the production of SVW-resistant hybrids [6,14,21]. However, recent reports describe the emergence of novel, less frequent races capable of overcoming these resistance sources [19].

Although the disease currently has a relatively low impact on yield in Argentina—likely due to the use of more resistant hybrids and no-tillage practices—it is endemic in the southern region of Buenos Aires, where the incidence of SVW has continued to rise [20]. Several factors highlight its potential as a latent threat, including the endemic nature of the pathogen, its broad host range among dicotyledonous species, and the long-lasting infectivity of its inoculum (microsclerotia), which increases soil inoculum loads under favorable weather conditions and susceptible plant hosts. Moreover, the observed diversity of Argentinian local races and their differentiation from those in other regions [22,23,24] underscore the need to gain further knowledge on the genetic architecture of resistance to identify both broad-spectrum and race-specific resistance sources for the development of improved varieties.

Despite the importance of SVW, the options for managing the disease are still limited. No-tillage practices have shown positive outcomes; however, since chemical control is ineffective for this pathosystem [17,25], genetic resistance remains the most reliable strategy for managing this pathogen. Although the identification of QTLs using biparental mapping populations and QTL pyramiding has enabled the development of commercial hybrids with SVW resistance, no sources of broad-spectrum resistance are yet available, and little is known about the genetic architecture of the trait. Exploring extensive germplasm sets and characterizing their behavior to SVW are the first steps toward identifying new sources of resistance that can contribute to sunflower breeding. In this vein, our previous studies have shown a wide range of responses to *V. dahliae* in two mapping populations (MPs) based on disease evaluations carried out under natural infection conditions [25]. Disease incidence (DI) and severity (DS) were measured in nine replicated field trials, with a total of 18 disease descriptors (DDs), covering phenological stages from R1 to R8. Multivariate analyses of disease descriptors revealed four tolerance groups within each mapping population. Notably, both populations showed similar averages for DI and DS despite different genetic backgrounds. Certain inbred lines in the INTA association mapping population (INTA-AMP) exhibited greater resistance than those in the biparental mapping population (BMP), whereas transgressive segregation in the BMP indicated successful recombination of genetic regions associated with resistance to SVW among the recombinant inbred lines (RILs) [25].

The availability of sunflower association mapping populations (SAM [26], AMP [27]) and high-quality genome references [28,29] enable genome-wide association studies (GWAS) to investigate the genetic architecture of resistance to SVW. Recently, Yu et al. [30] reported a GWAS study using the SAM population [26], in which 231 lines were inoculated under greenhouse conditions with a *V. dahliae* strain isolated in China. This study identified several candidate genes distributed across nearly all sunflower chromosomes, suggesting a polygenic architecture for *V. dahliae* resistance in sunflower [30]. However, no large-scale GWAS analysis has been conducted under natural infection conditions to date.

In this study, we employed ddRADseq to generate genome-wide single nucleotide polymorphism (SNP) data for the BMP, and integrated genetic and phenotypic data from Filippi et al. [31] and Montecchia et al. [25] to identify and validate QTLs associated with *V. dahliae* resistance using both association and biparental mapping approaches. These efforts aim to elucidate the genetic pathways and candidate genes involved in plant defense responses, thereby enhancing our understanding of SVW resistance under natural field conditions.

## 2. Materials and Methods

### 2.1. Mapping Populations and SVW Phenotypic Data

The BMP used in this study comprised 117 RILs (F7:11) derived from the cross between the inbred lines PAC2 and RHA439. PAC2 is a restorer line susceptible to SVW, whereas RHA439 is a highly tolerant restorer line. For GWAS, we used the Association Mapping Population of INTA (INTA-AMP), which consisted of 132 inbred lines, preserved at the Active Germplasm Bank of INTA Manfredi. The INTA-AMP was designed to balance genetic diversity [27,31,32,33] with adaptation to local growing conditions. In addition, these inbred lines showed high phenotypic variability in response to biotic and abiotic stresses (e.g., Sclerotinia Head Rot [34]; SVW [25]; Phomopsis Stem Canker [35]; Senescence [36]; and Drought tolerance [37,38]). For a more detailed genetic characterization of the INTA-AMP, see Filippi et al. [32].

The complete SVW phenotypic response data for both MPs can be found in Montecchia et al. [25]. Briefly, the MPs were evaluated at four phenological stages for disease incidence (DI), disease severity (DS), binomial disease severity (bDS), disease intensity (DInt), and the Area under the Disease Progress Curve (AUDPC) for DI and DInt, resulting in a total of 18 disease descriptors (DDs). The BMP was evaluated in four field trials (FTs) conducted at the EEA-INTA Balcarce, Balcarce, Buenos Aires (37°50′0″ S, 58°15′33″ W), during growing seasons 2013/14 to 2016/17. The AMP was evaluated in five FTs, four at EEA-INTA Balcarce (2014/15 to 2017/18) and a fifth (2017/2018) at “El Cencerro” seed company, Coronel Suárez, Buenos Aires (37°25′52.0″ S, 61°51′32.5″ W). In this study, we selected the five most informative DDs for linkage and association mapping analyses, based on their potential impact on yield components and the results obtained in Montecchia et al. [25]. The selected DDs were: DI at flowering (DI.Flw), DI.AUDPC, DS at grain filling (DS.Gf), bDS at grain filling (bDS.Gf), and PC1 from a principal component analysis which integrated the 18 DDs from Montecchia et al. [25]. Principal Component Analysis (PCA) successfully translated the high positive correlations observed between the 18 DDs to synthetic variables explaining a high proportion of the global phenotypic variance (PhV) observed in both MPs. For the AMP, the first two Principal Components explained 86.1% of the total variance (PC1: 80.3%; PC2: 5.83%) whereas for the BMP, they explained 91.1% of the multivariate variance (PC1: 85.1%; PC2: 6.1%). Both MPs were slightly skewed towards a higher proportion of moderately tolerant materials: 67% and 60% of the AMP and BMP were included within the tolerant and moderately tolerant clusters, respectively (Appendix A, Appendix A). The highly susceptible cluster comprised 12% and 8% of the AMP and BMP, respectively. The selected DDs showed moderate to high broad-sense heritabilities across field trials. Average heritabilities ranged from 34.28 to 53.2% in the AMP and from 24.01 to 43.87% in the BMP. As expected, based on their genetic diversity, the broad-sense heritability estimates for AMP were consistently higher than those of the BMP. The Best Linear Unbiased Predictors (BLUPs), obtained from mixed-model fitting across field trials for each trait and PC1, were used as response variables in all mapping analyses.

### 2.2. Genotypic Data

In this study, SNP data were generated for the BMP as described in the following section using double digest restriction-site-associated DNA sequencing (ddRADseq). For the AMP, we used the ddRADseq SNP matrix from Filippi et al. [31]. The original matrix of 18,161 SNPs with minor allele frequencies ≥ 0.05 was simplified to 9888 non-redundant SNPs by selecting a single representative SNP per linkage disequilibrium block. The chromosome distribution of these SNPs in the sunflower reference genome [31] is provided in Appendix A.

### 2.3. Genotyping of the BMP

Leaves from three to five plants from each IL were sampled in the field, immediately frozen in liquid nitrogen, and stored at −80 °C. Plant material was ground on a TissueLyser system (QIAGEN, Hilden, Germany), and DNA was extracted by using the NUCLEOSPIN Plant II kit (Macherey-Nagel, Düren, Germany) or the DNAeasy Plant Mini Kit (QIAGEN). ddRADseq genotyping was performed as described in Aguirre et al. [39] at the Genomics Unit of IABIMO, INTA-CONICET. Libraries were generated by digestion with the restriction enzymes SphI-HF and MboI. Size selection was performed automatically using SageELF^TM^ equipment (Sage Science, Beverly, MA, USA), with a fragment size of 450 ± 35 bp. Libraries were sequenced with an Illumina NextSeq 550 apparatus at the Virology Laboratory of the “Hospital de Niños Ricardo Gutiérrez”, Buenos Aires, Argentina. The resulting 75 bp paired-end reads were mapped to the reference genome (HanXRQ.v1, [28]) using Bowtie2 [40]. Samtools [41] was used to convert from SAM to BAM format and to sort the files. Subsequently, the ref_map module of Stacks [42] was used for SNP variant calling. The resulting SNP matrix was further refined using VCFtools [41]. Filtering parameters included position quality > 30, allele depth > 3 reads, minor allele frequency (MAF) > 0.05, and a maximum of 50% missing data following the criteria described in Filippi et al. [31]. Missing data were imputed using the imputation strategy proposed by Merino [43].

### 2.4. Genome-Wide Association Studies (GWAS)

The association between SNPs and phenotypes was assessed using statgenGWAS [44] in R [45], following the strategy outlined in Kang et al. [46]. To account for population structure, the Admixture software v1.3.0 [47] was used to identify genetic clusters using the complete set of non-redundant SNPs. The cross-validation analysis identified six as the most likely number of ancestry groups (K = 6, Appendix A). The estimated ancestry proportions for each inbred line (IL) were then used to create the Q matrix, which was incorporated into the GWAS models. Additionally, an identity-by-state unscaled kinship matrix (K) was estimated using the Van Raden [48] method and included in the model to control for more subtle population structure. Since sunflower hybrid production depends on the CMS/Rf (Cytoplasmic Male Sterility and fertility Restoration) system, a third matrix (CMS) reflecting the fertility status of each line was also added to control for additional sources of stratification. A single-locus Mixed Linear Model (MLM) GWA was fitted for each trait under different Q, K, and CMS matrix combinations. The first model included only the Q matrix, the second only the K matrix, the third both Q and K, the fourth K and CMS, the fifth Q and CMS, and the final model included all three matrices (Q + K + CMS). Additionally, the −log10 (1/n) threshold (where *n* is the total number of SNPs tested for association) [49] was applied to identify SNPs with statistically significant associations to the traits under investigation.

### 2.5. Composite Interval Mapping (CIM) on the Biparental Population

The linkage map was constructed using the R package onemap v3.2.0 [50], following a systematic approach for the organization and analysis of genetic markers. First, redundant markers among the 2292 polymorphic SNPs on the BMP were grouped into bins using the functions find_bins and create_data_bins. Subsequently, the function test_segregation was used to identify bins that deviated from the expected 1:1 segregation ratio, based on the chi-square test followed by a Bonferroni correction (α 0.05); those exhibiting distorted segregation were excluded from the analysis. Next, two-point mapping was performed by applying a LOD threshold of 6 and setting a maximum recombination frequency of 0.5. The resulting linkage groups were defined based on the reference genome HanXRQ.v1 [28]. Recombination distances were calculated using Kosambi’s mapping function [51], while the marker order within each group was determined using the record function [52].

CIM was performed using the linkage map and the five SVW disease descriptors, employing the R-qtl package [53]. The *cim* function automatically selected up to three cofactors outside the sliding window of 20 cM, where the tests were sequentially carried out for each single trait. Using the *scanone* function, a Haley–Knott regression test with 5000 permutations was conducted to obtain a genome-wide LOD significance threshold of *p* < 0.01 and *p* < 0.05 for each trait. Given that the significance threshold LOD scores showed small variation between traits, the most astringent ones were selected as thresholds for the five variables. The procedure for detecting multiple significant QTLs and adjusting a linear model to estimate their effects on each phenotypic variable followed the methods described by Broman [54]. After adjusting a multiple-QTL fixed model, the QTL positions were refined by maximum likelihood by the function *refineqtl*. Based on these positions, the closest SNPs to the QTL peak were defined.

### 2.6. SNP Matrices Alignment to New Versions of Sunflower Reference Genomes

To determine the positions of the SNPs in the HanXRQ.v2.0 and HA412-HO.v2.0 assemblies, we extracted windows of 70, 100, 200, 500, 1000, and 5000 bps sequences around the SNP positions in HanXRQ.v1. Then, we mapped these sets of extracted sequences with different lengths to each assembly using Bowtie 2, ensuring no mismatches (--very-sensitive -N 0 --score-min L,0,0 in end-to-end mode). We defined the final position of an SNP in the new version assembly using custom scripts. We searched for a minimum fragment length that provides an exact and unique alignment between the fragment (centered on the SNP) and the assembly. We then confirmed this position using fragments of greater length. For example, if we found a fragment of 100 bp that mapped exactly and uniquely, we used the alignments of fragments of 200, 500, 1000, and 5000 bp to confirm the obtained position. Due to the genetic distance between the HanXRQ and HA412-HO genotypes, some SNPs could not be assigned to a position using exact alignments. For these SNPs, we re-mapped all fragments of different sizes, allowing mismatches (--very-fast end-to-end mode), and proceeded in the same way, selecting the minimum fragment size with a unique alignment to define the position in the assembly.

### 2.7. Candidate Gene (CG) Analysis

To identify CGs near the associated SNPs, we analyzed 400 Kbp windows centered at the SNP position using the intersect command from the BEDTools package [55] on the HanXRQ.v2 structural annotation. We updated the functional annotation of the corresponding proteins using InterProScan v.5.66-98.0 [56]. A comprehensive overview of the putative functions of the genes within the window of 400 Kbp around the associated SNP position was generated using different approaches. First, we searched for annotations related to defense functions. Therefore, we also compared our list of genes with modules related to defense in the reference gene networks reported by Ribone et al. [57] to detect new CGs. Next, we compiled the GO Terms of the different CGs and summarized them with REVIGO [58]. Finally, we re-annotated the genes with DRAGO3 to predict disease resistance genes (R genes).

## 3. Results

### 3.1. GWAS Analysis for SVW Resistance

Comparison of the tested MLMs revealed that the Q+K model best fits the data for the five traits. The corresponding Q-Q plots and genomic inflation factors are presented in Appendix A [1], respectively. The average inflation factor across traits for this model was 1.013 which is very close to the theoretical = 1 [47]. The LOD threshold established for significance was 3.995.

A total of 16 statistically significant marker-trait associations were identified for SVW resistance, involving nine SNPs distributed across three chromosomes (CHR08, CHR09, and CHR10 of HanXRQ.v1), with several SNPs associated with more than one DD (Table 1, Figure 1 and Appendix A). Among the nine associated SNPs, five were exclusively linked to DI.AUDPC, the trait with the highest number of statistically significant associations (eight SNPs). Notably, five of these SNPs were clustered within a 12.8 Mb region on CHR10, while the remaining three were in two regions of CHR08 and one region of CHR09. The PhV of DI.AUDPC explained by these regions was 0.306 for CHR10, 0.131 and 0.156 for the two regions on CHR08, and 0.140 for the region on CHR09 (Table 1). These results highlight the prominent role of CHR10 in disease control, as 12 out of the 16 associations corresponded to SNPs mapped on this chromosome. Remarkably, SNP 10511, located in the middle of the previously mentioned 12.8 Mb region, was associated with all five evaluated DDs and ranked first in all except bDS.Gf, where it ranked second following SNP 6824. Moreover, SNP 10511 explained the largest variance proportions for all traits (DI.Flw = 0.282; DI.AUDPC = 0.307; DS.Gf = 0.270; bDS.Gf = 0.262; and PC1 = 0.294).

Two additional markers on CHR10, SNP 9300 and SNP 10561, were jointly associated with DI.Flw, explaining 0.142 and 0.239 of the PhV for this trait, respectively. Individually, SNP 9300 was also associated with DS.Gf, and explained 0.162 of the PhV, while SNP 10561 was associated with DI.AUDPC, accounting for 0.263 of the total variance for this trait. SNP 9300 is located on the proximal region of CHR10, and SNP 10561 is positioned on the distal region, 7.3 Mb downstream from SNP 10511. The marker on CHR08, SNP 6824, was the most significant for bDS.Gf and ranked fifth for DI.AUDPC. This SNP also showed a nearly significant LOD score of 3.952 for DI.Flw (LOD-threshold = 3.995), suggesting a potential role for CHR08 in DI.Flw as well. Overall, these results emphasize the oligo-genic nature of SVW resistance in sunflower under natural infection conditions, with a large effect region on CHR10 (Figure 1). Considering the cumulative effect of the genomic regions associated with each trait, instead of the isolated SNPs, the proportions of PhV explained by these GWAS results were 0.424 for DI.Flw, 0.604 for DI.AUDPC, 0.431 for DS.Gf, 0.405 for bDS.Gf, and 0.294 for PC1 (Table 1).

### 3.2. Genotyping and Composite Interval Mapping on the BMP

The Next-Seq run yielded an average of approximately 1.5 million reads per sample and the proportion of missing data was about 59.9%. A total of 6294 SNPs were detected in both parental lines and the progeny by aligning the reads to HanXRQ.v1. Given the high proportion of missing data and their variation across samples, an ad hoc imputation pipeline was designed and deployed for imputing BMP’s missing data. This data imputation strategy yielded a complete matrix of 3561 SNPs screened both in parental and in progeny lines, with a MAF > 0.05. Of this set, 2292 SNPs could be mapped to the reference genome and were polymorphic between the parental lines (Appendix A). To remove redundant SNPs, 1976 bins were created and out of them 1769 that fit the expected segregation ratio were used for map construction. The linkage map obtained consisted of 17 linkage groups (LGs) with an average synteny of 91.33% and an average collinearity with the reference genome of 79.69%. The average SNP density across LGs was 0.438 SNPs/cM, with LG17 being the LG with the highest coverage with 11.19% of total SNPs and LG 16 the one with the lowest coverage, with 1.75% of total SNPs (Appendix A). The representation of the SNPs distribution on the genetic map and its collinearity with the reference genome (HanXRQ.v1) are shown in Appendix A.

A total of 17 QTLs were identified by the CIM approach for SVW resistance distributed across CHR10, CHR17, CHR14, CHR08, and CHR13. CIM analysis determined LOD values of 4.15 and 3.34 as significance thresholds of α 0.01 and α 0.05, respectively. Twelve QTLs were significant with a 1% threshold and five QTLs with a 5% threshold. QTLs on CHR10 showed the highest LOD scores and the greatest impact on PhV for all traits. The numbers of QTLs detected per trait and chromosome were four, four, two, three, and four for DI.Flw, DI.AUDPC, DS.Gf, bDS.Gf, and PC1, respectively (Figure 2).

By comparing the QTL peak positions, their confidence intervals (CIs, defined by a −1 LOD drop region around the peak on the genetic map), and the physical positions of the SNPs closest to the QTL peaks, we found that most of the detected QTLs were shared across different traits, except for two: one on CHR13 (PC1) and another on CHR08 (DI.AUDPC) (Table 2). Remarkably, the QTLs found on CHR10 and CHR17 were associated with all traits at the highest significance level. The QTL identified on CHR10, centered at 58 cM (*qSVW-10.1*), was the most significant for all traits, with LOD scores of 7.80, 7.74, 5.62, 7.33, and 7.06 for DI.Flw, DI.AUDPC, DS.Gf, bDS.Gf, and PC1, respectively. The proportion of PhV explained by this region was also the highest for the five traits, ranging from 15.24% (DS.Gf) to 20.28% (DI.AUDPC), with an average of 18.8% across traits. The QTL found in CHR17, centered at 259 cM (*qSVW-17.1*), was also associated with all traits at the highest significance threshold. The proportion of PhV explained by this region was the second highest for the five traits, ranging from 11.27% (DS.Gf) to 18.73% (bDS.Gf). The mean phenotypic proportion explained by this QTL across all traits was 14.13% (Table 2).

The QTL identified on CHR14, centered at 119 cM (*qSVW-14.1*), was associated with four traits, DI.Flw (α 0.01), DI.AUDPC, bDS.Gf, and PC1 (α 0.05), and the PhV explained by this region ranged from 7.47% (PC1) to 12.89% (DI.Flw) with an average proportion of 9.38% across traits.

The QTLs located on CHR08 cover two different regions linked to two DI-related traits, one centered on 220 cM (*qSVW-08.1*: DI.Flw) and the other on 264 cM (*qSVW-08.2:* DI.AUDPC). The closest markers to each peak on CHR08 are separated by 11.95 Mb and their CIs, although overlapping, differ significantly in their span. The proportion of PhV explained by *qSVW-08.1* for DI.Flw was 9.44%, while *qSVW-08.2* explained 10.82% for DI.AUDPC. Finally, a QTL on CHR13, centered on 53 cM (*qSVW-13.1*), was identified for PC1, which explained 6.33% of its variance.

In sum, six QTLs distributed on five chromosomes were highly involved in SVW resistance. In five out of the six QTLs identified in the BMP, the parental allele derived from RHA439 reduced DI and DS, thus showing additive effects of negative sign and increased resistance to SVW. The only QTL that showed an additive effect of positive sign, derived from PAC2, was designated *qSVW-13.1* (PC1). The linear regression models that included the multiple QTLs identified for each trait explained proportions of PhV of 52.42%, 51.48%, 26.16%, 44.97%, and 47.47% for DI.Flw, DI.AUDPC, DS.Gf, bDS.Gf, and PC1, respectively.

The putative physical locations of each QTL on the reference genome (HanXRQ.v1) were approximated using the peak’s closest SNP position, plus the genetic intervals defined by the most frequent flanking markers on either side of the peak (Table 2). The intervals between flanking markers, defined in centiMorgans (cM), comprehended sets of SNPs that could be referred to genomic regions by their physical positions. The resolution of these intervals varied along the different QTLs; it was found that *qSVW-10.1* ranges from 48 to 78.73 cM (interval = 30.73 cM), *qSVW-17.1* from 255 to 262.6 cM (interval = 7.35 cm), *qSVW-14.1* ranges from 115.36 to 123.01 cM (interval = 7.65 cM), *qSVW-08.1* ranges widely from 117.84 to 270.69 cM (interval = 152.84 cM) and *qSVW-08.2* ranges from 251.68 to 270.69 cM (interval = 19.01 cM); finally, *qSVW-13.1* ranges from 15.23 to 114.12 cM (interval = 98.89 cM) (Table 2 and Appendix A).

Focusing on CHR10, due to its significant role in disease control, we identified 14 SNPs within a 30.73 cM interval. When sorted by physical position, these SNPs spanned a 20.91 Mb region, with a core set of 11 SNPs concentrated within a 7.5 Mb segment (201.95–209.45 Mb, Appendix A), which is contained within the 12.8 Mb region detected by GWAS. It is noteworthy that all these SNPs have significant LOD values, beyond the—1 LOD decay, given the high significance level reached by this region in all traits.

### 3.3. Cross-Reference Assignment of Associated SNPs and Mining of Candidate Genes

In this study, SNPs obtained for both MPs were initially referenced to HanXRQ.v1 [28]. However, with the release of an updated genome assembly and annotation in 2020 (HanXRQ.v2), along with the more recent HA412-HO assembly [29], we re-mapped the SNPs associated with SVW to update the physical positions of the associated SNPs (Appendix A).

All 18 associated SNPs (nine SNPs from the GWA and nine from the BMP mapping approach) were found in the HanXRQ.v2 and HA412-HO.v2 genomes. In the HanXRQ.v2. genome, most SNPs remained in the same chromosomes, except for SNP 9300, located in the proximal region of CHR10 (59.1 Mb) in HanXRQ.v1 and in the distal region of CHR08 (154.4 Mb) in HanXRQ.v2, and SNP 6824, located on CHR08 (19.4 Mb) in HanXRQ.v1 and on CHR11 (60.5 Mb) in HanXRQ.v2. In CHR11, no associations were found previously (Appendix A). In the BMP, SNPs were only slightly re-located to different positions within the same chromosomes of the HanXRQ.v2 assembly, maintaining their relative distances in most chromosomes (Appendix A). Results of the mapping of AMP SNPs on the HA412-HO.v2 reference were less congruent than those observed for HanXRQ.v2. Only five out of nine SNPs were mapped to the same chromosomes as in HanXRQ.v1 and two of these were mapped in more than one region. Two SNPs located on the large effect region of CHR10 in HanXRQ.v1 shifted to different chromosomes in HA412-HO.v2 (Appendix A). The BMP markers were more congruent between genomes, with only one chromosome shift, which also mapped to more than one region in HA412-HO.v2 (Appendix A).

Most associated SNPs were found on the distal arm of CHR10. In the GWA study, five out of nine SNPs encompass a region of 12.85 Mb in HanXRQ.v1 (196.3–209.2 Mb), centered around SNP 10511. The re-localizations in XRQ.v2 displace this window (151.9–159.8 Mb) and position the associated SNP 10452 towards the center of this window, next to SNP 10511, narrowing this region down to 7.96 Mb (a reduction of 4.8 Mb). Additionally, the distance between the SNPs with the largest number of associated traits, 10511 and 10561, comprises only 5.3 Mb, reinforcing the relevance of this region on SVW resistance control. Remarkably, three associated SNPs of the biparental mapping study (SNPs 86706_53, 86780_47, and 86923_8) are found near this region. These three SNPs encompass 3.45 Mb (HanXRQ.v2) and overlap with the GWA region over a 1.07 Mb window (Figure 3 and Appendix A). Overall, the eight SNPs associated by both methods on CHR10 encompass a region of 10.34 Mb (Figure 3).

Both mapping methods also found significant associations on CHR08. The re-localizations in XRQ.v2 position two SNPs (7394 and 9300) of the GWA study in distinct regions, while two SNPs (63166_57 and 63838_12) of the CIM study are 12.8 Mb apart. These results emphasize the effect of CHR08 on SVW resistance, with an average of 14.4% of PhV explained for the AMP traits and smaller effects on the BMP (average 10.1%).

The QTL study detected another interesting region in CHR17, where two close SNPs (159430_35 and 159418_28) were associated with one and four traits, respectively. Interestingly, all traits were associated by both SNPs in the QTL *qSVW-17.1*, explaining considerable proportions of PhV, from 11.27 to 18.73%. However, this genomic region was not detected by the GWA performed here.

### 3.4. Candidate Genes

Given the significant improvements in assembly and annotation in HanXRQ.v2 compared to HanXRQ.v1, candidate gene mining was performed using the most recent reference genome. By analyzing a 400,000 bp region surrounding each associated SNP in the HanXRQv2.0 genome, we identified 109 candidate genes (CGs) in the GWA study and 99 CGs in the CIM study. Due to the proximity of two SNPs in the GWA study, only 96 unique genes were identified in this analysis, resulting in a total of 195 unique candidate genes across both mapping methods (Table 3).

A reannotation of the CGs’ function revealed very similar results to those of the HanXRQv2.0 genome (Appendix A). The putative function of the different genes is diverse, including transmembrane receptors, kinases, genes related to metabolism, among many others. To detect genes putatively related to defense processes, we compared our 195 CGs to genes of modules enriched in defense functions to fungal pathogens in sunflower co-expression networks [57]. Two genes were found in module “Green16” from Ribone et al. [57]: HanXRQr2_Chr10g0456461 (“glutamine synthetase, chloroplastic”) and HanXRQr2_Chr11g0492861 (“omega-3 fatty acid desaturase, endoplasmic reticulum-like”), suggesting a possible role of these genes in the defense response to *V. dahliae*.

To interpret the functional annotation data of the different CGs, the corresponding GO Terms were summarized with REVIGO [58] to detect the most frequent and relevant of them (Appendix A). The most frequent GO Terms in both mapping methods were “membrane” (CC, closely followed by “cytoplasm” in this category), “nucleic acid binding” (MF, closely followed by “metal ion binding” and “ATP binding” in this category) and “transmembrane transport” (BP, followed by “regulation of DNA-templated transcription”, “signal transduction”, and “proteolysis” in this category). However, over 68% of the terms identified in the CGs by each mapping method were unique to that method. Among the CGs of the GWA study, other relevant frequent terms are “lipid metabolic process” and “transmembrane transporter activity”. Instead, among the CGs of the CIM study, the most frequent and relevant terms were “catalytic activity”, “hydrolase activity”, and “carboxylic acid metabolic process”. Terms related to “defense” were not found.

Additionally, DRAGO3 was used to annotate CGs and identify novel resistance genes. Of the 195 CGs, 11 were classified as putative resistance genes, with nine located within the 10.34 Mb resistance region detected on CHR 10 (Table 4, Figure 3).

Since eight of the 18 associated SNPs were located within a 10.34 Mb region on CHR10, we conducted a more exhaustive search within this region to identify additional CGs (Figure 3). The 356 CGs identified in this 10.34 Mb section were re-annotated with DRAGO3, and 25 putative R genes were detected. Moreover, 13 of them were located within a 320 kb window close to SNPs 10511, which was significantly associated with all traits, and 10513. We also searched for CGs with GO Terms related to “defense”, finding a total of seven genes with GO Terms “regulation of defense response to fungus, incompatible interaction”, “defense response to fungus”, “regulation of defense response”, and “defense response”. Finally, to identify genes putatively related to defense processes, but with no such annotation, we searched in the modules enriched in defense functions to fungal pathogens of Ribone et al. [57] finding seven additional CGs in modules “Green5” (HanXRQr2_Chr10g0455491, HanXRQr2_Chr10g0455961), “Green 16” (HanXRQr2_Chr10g0456461, HanXRQr2_Chr10g0456071), “Green24” (HanXRQr2_Chr10g0455561), “Root18” (HanXRQr2_Chr10g0456361), “Root36” (HanXRQr2_Chr10g0454531), and “Root44” (HanXRQr2_Chr10g0455491). In summary, 39 of the CGs examined within the resistance region of CHR10 have annotations and/or expression patterns supporting their role in defense responses.

## 4. Discussion

SVW is a well-established disease in Argentina, with a long-standing endemic prevalence in the sunflower growing region, particularly in southern Buenos Aires Province and eastern La Pampa. The existence of pathogenic local races of *V. dahliae*, genetically diverse from those of other regions [14,19,22,23], increases the complexity of disentangling the genetic architecture of SVW resistance. Thus, a deeper understanding of broad-spectrum resistance [9,10,11,25] complementing race-specific studies [4,21,24] is highly necessary for the development of SVW-resistant materials. This paper presents an integrated study of SVW resistance, based on large-scale GWAS and CIM approaches, using a thorough phenotypic characterization of two MPs under field growing conditions, in highly infested inoculum reservoirs. The infested fields where the MPs were tested harbor extreme inoculum loads and a wide diversity of *V. dahliae* isolates, sustained by sunflower monocropping for more than two decades, by the time these studies were conducted [25,59].

The overall analysis of GWAS-identified associations revealed a consistent behavior among traits and SNPs. Comparing the results from different GWAS MLM models tested on the AMP highlighted the importance of accounting for population structure to control Type-I error in association models, while the fertility status of inbred lines (CMS) had minimal impact (Appendix A). The inflation factors for the five traits studied were close to the expected theoretical value, supporting the reliability of these results [46]. With the updated marker positions in HanXRQ.v2, disease resistance was associated with only four chromosomes, with the associations explaining substantial proportions of phenotypic variance across all traits. Notably, four of the nine associated SNPs exhibited strong signals for multiple traits. Additionally, the presence of five associated SNPs within a relatively small region on chromosome 10 (196.3–209.2 Mb in HanXRQ.v1) highlights the importance of this genomic region in the response to SVW. The pattern of associations observed across the five evaluated traits reveals two key aspects of SVW resistance. First, the most significant associations and the largest effects on phenotypic variance are concentrated on the distal arm of CHR10. Second, the distribution of the nine significantly associated SNPs across four chromosomes, in relatively small genomic regions, points to the oligo-genic nature of SVW resistance.

In contrast with these results, in a recent GWA study Yu et al. [30] reported a polygenic architecture of SVW resistance. They conducted an artificially inoculated trial assessing the response to SVW of 231 inbreds from the SAM diversity panel [26] under greenhouse growing conditions. A single *V. dahliae* strain isolated from Wuyuan, China was used as an inoculum. They identified 148 QTLs across all chromosomes and highlighted 16 QTLs bearing 23 CGs related to disease resistance. These genes were distributed on eight chromosomes (excluding CHR10 and CHR14) and the authors underlined CHR04, CHR05, CHR09, and CHR011. Surprisingly, none of the 23 CGs stressed by this study mapped close to any of our associated SNPs or regions (considering the same reference genome) within those eight chromosomes. Unexpectedly, in the light of our results, Yu et al. [30] did not find any highly significant associations or gene clusters on CHR10. Their conclusion of a polygenic architecture for this trait collides significantly with our findings. However, several aspects need to be taken into consideration to allow comparison between both studies. The AMP has 21% and 12% of susceptible and highly susceptible inbreds, respectively, under these evaluation conditions, while Yu et al. [30] reported that only eight out of 231 materials from their panel were susceptible, and none was scored as highly susceptible to the *V. dahliae* strain inoculated. This represents only 3.46% of the lines within their panel. This underrepresentation of susceptible and highly susceptible materials might have led to limitations in the analysis for detecting QTLs of major effects controlling SVW. On the other hand, in our phenotyping studies we exposed our MPs to an extreme disease pressure under field growing conditions, in an inoculum reservoir known to contain *V. dahliae* strains collected from the whole sunflower-growing region of the Southern Pampas [25], where different races have been reported [13,14,19]. Thus, the association profile revealed by our studies would correspond to genetic factors harbored in the AMP contributing to SVW broad-spectrum resistance to the different *V. dahliae* local strains, with an oligo-genic configuration.

For the CIM analysis of the biparental population, the matrix of genotypic data was obtained using the ddRADseq method, with the restriction enzyme pair of SphI-HF/MboI. The performance of this method was as expected [60,61]. The constructed linkage map showed high levels of synteny and collinearity with the reference genome across linkage groups (LGs), with averages of 79.69% and 91.33%, respectively (Appendix A). These results support the map’s reliability for QTL mapping [62]. Consistent with the GWAS findings, disease response was associated with only five chromosomes (10, 17, 14, 8, and 13), with three of the six QTLs detected influencing multiple traits. Among these, *qSVW-10.1* and *qSVW-17.1* were the most prominent, as they were associated with all five traits and explained an average of 18.8% and 14.13% of the phenotypic variance, respectively. All QTLs, except for the one on CHR13, had negative additive effects, indicating that the favorable alleles for these regions originated from RHA439. As in the AMP, CHR10 (*qSVW-10.1*) played a dominant role among the associated regions, though CHR17 (*qSVW-17.1*) also contributed significantly to SVW response in the BMP. It is noteworthy that while our single-locus GWAS approach did not detect any significant SNPs on chromosome 17, other methods did. For instance, the mixed model with multiple loci [63] identified significant SNPs for DI.Flw and PC1 on this chromosome, although they explained only small portions of the phenotypic variance for these traits [24]. Remarkably, the genomic regions of chromosome 10 identified by both GWAS and CIM were almost identical.

The overlap between GWAS and QTL mapping results on the distal arm of CHR10 helped narrow the region associated with disease control. The confidence interval (CI) for *qSVW-10.1* spans from 190.6 to 211.5 Mb (HanXRQ.v1), with key SNPs at 207.5 Mb (SNP 86706_53, DI.Flw), 208.8 Mb (SNP 86780_47, DI.AUDPC, DS.Gf, and PC1), and 211.29 Mb (SNP 86923_8, bDS.Gf), which correspond to QTL peaks. Nine additional significant SNPs from the CI are located between GWAS SNPs 10511 (201.9 Mb) and 10561 (209.21 Mb). Notably, SNP 10561 in the AMP is flanked by SNP 86780_47 (4.11 kb upstream) and SNP 86815_34 (2.43 kb downstream) (Figure 3). This region, spanning from SNP 10511 (201.9 Mb) to SNP 86815_34 (209.45 Mb), was independently detected by both mapping approaches, covering 7.55 Mb within the broader 12.8 Mb window on HanXRQ.v1 (which covered 10.34 Mb in HanXRQ.v2). These findings confirm the high synteny and collinearity of the linkage map with the reference genome, underscoring the major role of this region in SVW broad-spectrum resistance across all resistance traits.

The results reported by the present study, where different mapping methods consistently converge on overlapping genomic regions on CHR10, with major effects on SVW resistance, are congruent to previous findings from the classical breeding approaches reporting a “single gene” driving SVW resistance [9,11,12]. Studies conducted on natural-infested field conditions in Argentina by Aggarwal et al. [64] on F3 and F4 families from a contrasting cross for SVW resistance also detected highly significant regions on CHR10 explaining up to 65% of PhV and signals of smaller effect on CHR11. Although Aggarwal et al. [64] used a smaller set of markers (188 SNP) and individuals at less advanced generations for mapping the trait, the significance of their results reinforces ours in similar growing conditions. The preponderance of CHR10 on SVW resistance to Argentinean isolates seems to be indisputable and the contribution, whether on bolstering or on race-specificity role, of the other QTLs detected by our study remains to be studied and disentangled.

SVW resistance was first reported as a qualitative trait governed by a single dominant locus [9]. Later, surpassing reports have described different materials with dominant, additive, and recessive sources of resistance to North American races [11,12]. Since then, the V1 locus detected by classical breeding was identified on the maintainer IL HA89 [9] and became the main source of resistance for hybrid development worldwide [10] in Argentina and later in the USA and Europe [4,10,13,15,16]. In Argentina, VArg1 and VArg2 local races were discovered among the isolates affecting sunflower and differential lines bearing race-specific resistance were reported [13,14] as well as less-frequent new races overcoming these resistance sources [19]. The molecular characterization of *V. dahliae* races isolated from sunflower, in comparison with reports from well-characterized pathosystems such as tomato [65,66,67] revealed the absence of effectors like Ave1 in Argentinean [24] and European sunflower isolates [4]. The identification of race-specific public differential lines and the study of the pathogenicity of European and Argentinean isolates over them shed light onto the complexity of this pathosystem from a global perspective [4]. A recent report characterized *V. dahliae* sunflower isolates from Argentina’s main growing region compared to European and North American strains, showing no geographical clustering at local level for molecularly different strains. This study found significant molecular relatedness between Argentinean and French isolates [23].

Mapping the associated SNPs to the reference genome HanXRQ.v2 [28] to improve candidate gene annotations revealed differences in the chromosomal positions of the associated SNPs between different versions of the reference assemblies. GWAS-associated SNPs were more affected by these differences than those identified by CIM. One of the most relevant changes is the localization of the QTL associated to SNP 6824 on CHR11, which was previously positioned on CHR08. Although Yu et al. [30] and Aggarwal et al. [65] reported candidate genes and a QTL for SVW resistance on CHR11, respectively, these regions were distant from those identified here. In addition, the interval of the major QTL for SVW in CHR10 was reduced from 12.8 Mb (HanXRQ.v1) to 7.96 Mb (HanXRQ.v2). On the other hand, the assignment of the same SNPs to the reference genome HA412-HO.v2 led to its localization on other chromosomes, potentially disrupting haplotype continuity. Such variations in the reference genome can therefore influence the identification of SNPs in linkage disequilibrium with the trait’s segregation.

Regarding candidate gene search near the 18 associated SNPs detected by both mapping approaches, a total of 195 unique genes were identified. Although none of these genes had a GO annotation directly related to defense, we were able to re-annotate 11 of them as putative R genes using DRAGO3. These R genes could be associated with immune responses, thus suggesting a putative role in the activation of effector-triggered immunity (ETI) [68] and downstream defense activation [69,70]. In addition, the most frequent GO terms found in both mapping populations, although corresponding to various molecular functions and biological processes, are also related to defense responses. They include locations of the frontline defense, such as “membrane” and “cytoplasm” (with “plasma membrane” less frequently noted), as well as processes involved in defense activation and signaling molecule production, such as “transmembrane transport”, “regulation of DNA-templated transcription”, “signal transduction”, and “nucleic acid binding”. Yu et al. [30] also identified many CGs as receptors or signaling components in the same pathosystem, though under different conditions.

Interestingly, the genes identified near SNPs from the GWA and CIM methods shared less than 35% of their GO terms, reflecting the distinct SNP associations observed across different chromosomes. Notably, frequent biological processes influenced by the genes near the associated SNPs in the GWA method were primarily linked to lipid metabolic processes, while those in the CIM method were more frequently associated with carboxylic acid metabolic processes. Further supporting potential shifts in metabolism, one gene from each mapping method was found in a co-expression gene network module (“Green16”), which is related to defense and enriched in metabolic genes [57]. The gene identified by CIM near SNP 86923_8, HanXRQr2_Chr10g0456461, is annotated as a “glutamine synthetase”, a key enzyme in plant glutamate metabolism, which plays a critical role in defense against pathogens [71]. The gene identified by GWA, HanXRQr2_Chr11g0492861, located near SNP 6824 in CHR11, is annotated as “omega-3 fatty acid desaturase,” involved in lipid remodeling and jasmonic acid (JA) production [72,73]. Notably, both SNPs, despite resulting from different mapping methods, are associated with the bDS.Gf trait.

We found and analyzed 356 genes located in the key region on CHR10. A total of seven genes were related to defense based on their GO terms, and six of them were also linked to lipid metabolic processes, reinforcing the possible role of lipid metabolism in the defense response. The alteration of lipid metabolism during pathogen infection has implications for the rearrangement of physical barriers (i.e., cuticle and plasma membrane), the expression of signaling molecules and antimicrobial compounds, reactive oxygen species (ROS) metabolism, and more [74,75]. Consistently, four of these defense-related genes are critical enzymes for the biosynthesis of plastoquinones and tocopherol, important antioxidants that protect against ROS [76,77]. Additionally, one gene is annotated as a diacylglycerol kinase 7-like, a key enzyme in plant lipid signaling [78]. Furthermore, the closest defense-related gene to SNP 10511, HanXRQr2_Chr10g0453441, is annotated as a long-chain acyl-CoA synthetase 2. This enzyme is involved in the biosynthesis of long-chain fatty acids, which are precursors of cutin and suberin, lipid polymers that constitute the cell wall of tissues like the epidermis, endodermis, and periderm in roots [79]. The root tips and sites of lateral root formation are the points of entrance for *V. dahliae*, since they lack the endodermis, which facilitates the pathogen’s access to the xylem [80]. These annotated functions (LACS2), plus another one related to lateral root formation [81], might contribute to the resistance to a soil-borne fungal pathogen that finds its principal barrier in endodermis. Finally, a gene associated with defense by Ribone et al. [57], HanXRQr2_Chr10g0455491, is annotated as a lipase-like PAD4. This lipase is responsible for breaking down triacylglycerols into smaller compounds (i.e., fatty acids, glycerol, and diacylglycerol), which are involved in cellular signaling [82,83]. In summary, genes involved in pre-formed structures that prevent the entrance of the pathogen, as well as those that induced defense mechanisms elicited by the presence of the fungus, could contribute to a resistant phenotype.

To reinforce the importance of this region in defense, 25 putative R genes were identified, nine of which were already found in the 400 kb window surrounding the SNPs 10484, 10511, 10513, 10452, and 86706_53. Notably, 13 out of 25 R genes were concentrated within a 320 kb window near SNPs 10511 and 10513. The number of R genes in an area associated by both mapping methods strongly suggests the activation of ETI. Overall, this region presents a significant number of candidate genes, 39 of which could be directly associated with plant immunity.

## 5. Conclusions

The study successfully combined QTL mapping methodologies across two independent populations, revealing a key genomic region on the distal arm of chromosome 10 (151.9–162.2 Mb HanXRQ.v2) that explains significant phenotypic variance proportions in all SVW resistance traits. Additional associated genomic regions were found on seven chromosomes, two identified exclusively by GWA, three identified exclusively by CIM and one chromosome identified by both methods, supporting the oligo-genic nature of SVW resistance. Considering the conditions under which our MPs were challenged by SVW, grown in inoculum reservoirs harboring a diverse set of *V. dahliae* strains, the identified genomic regions can be considered crucial for SVW broad-spectrum resistance and might also play a significant role on race-specific resistance. Furthermore, the genes identified near the associated SNPs suggest the activation of ETI and associated metabolic changes that may lead to the synthesis of signaling molecules, including plant hormones. This implies a defensive response that could involve the restructuring of the root cell wall at the infection site. Of particular interest are the 39 candidate genes located within the major QTL detected on CHR10, which are likely involved in both preexisting defense mechanisms and the immune response triggered by pathogen invasion. Special attention should be given to this region for further exploration of potential targets for enhancing disease resistance.

Finally, the results of this work provide valuable resources for plant breeding purposes and to deepen the research towards race-specific studies and plant–pathogen interactions.

## Figures and Tables

**Figure 1 plants-14-01187-f001:**
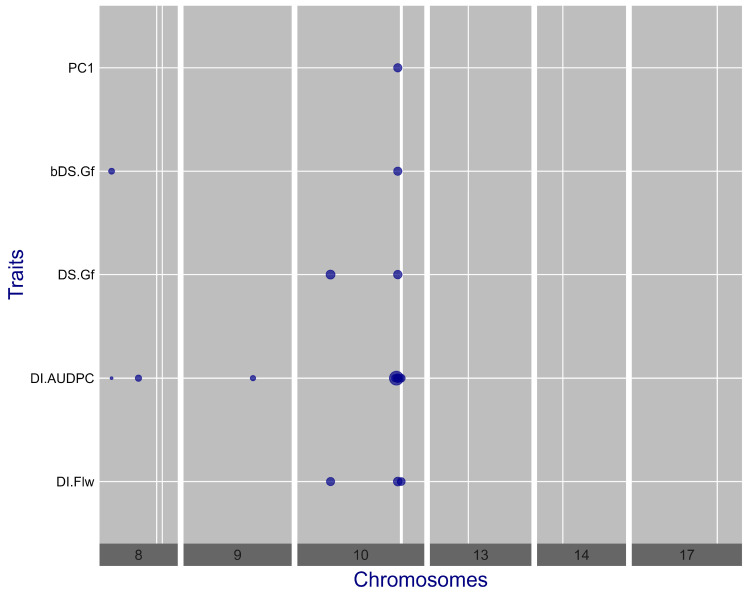
Plots of effect sizes for the associated markers to five SVW disease descriptors (DI.Flw, DI.AUDPC, DS.Gf, bDS.Gf, and PC1). Dot diameters are proportional to the effect size of the associated marker. Thinner vertical lines indicate the genomic positions of the closest SNPs to the SVW-resistance QTL’s peaks associated by Composite Interval Mapping analysis.

**Figure 2 plants-14-01187-f002:**
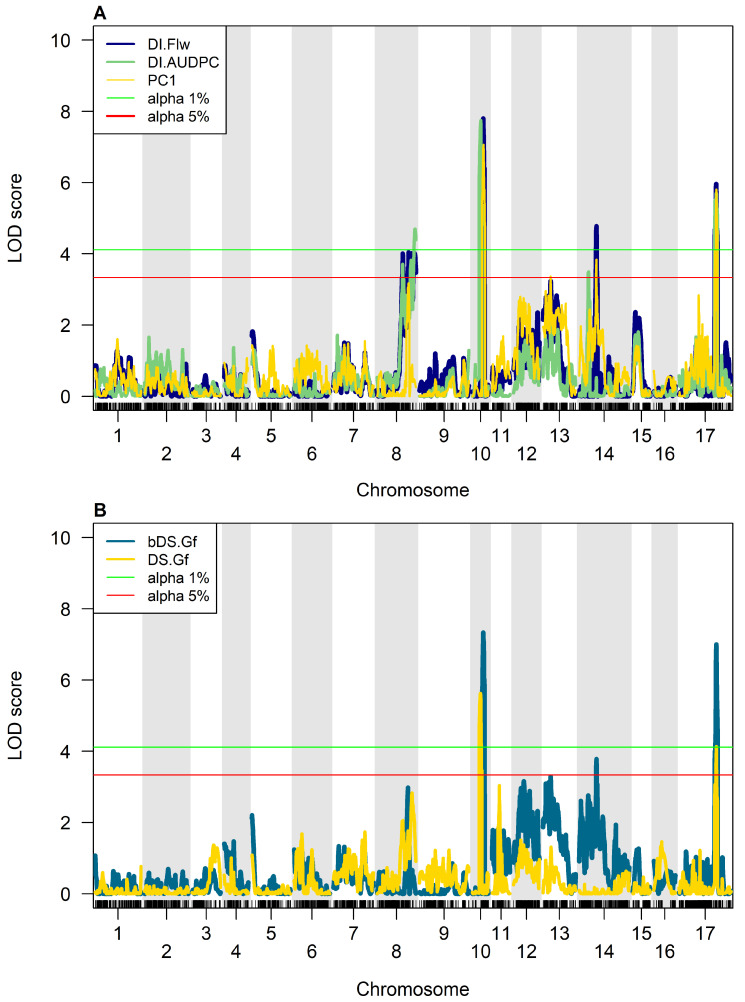
SVW Resistance QTLs identified through Composite Interval Mapping analysis on BMP: (**A**) QTLs identified for the disease descriptors related to Disease Incidence (DI.Flw, DI.AUDPC, PC1). (**B**) QTLs identified for the disease descriptors related to Disease Severity (bDS.Gf, DS.Gf). Red and green lines represent the significance statistical thresholds at *p*  <  0.05 and *p*  <  0.01, respectively.

**Figure 3 plants-14-01187-f003:**
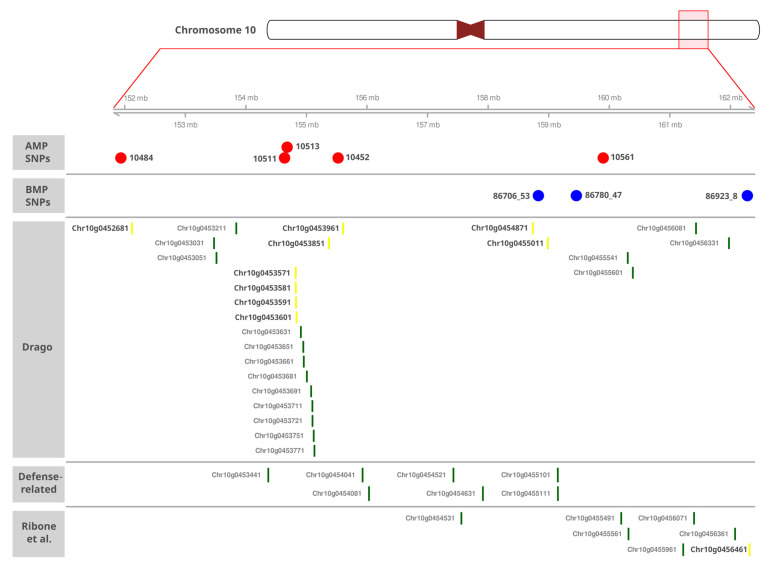
Region of 10.34 Mb corresponding to the major QTL for SVW resistance detected on CHR 10. Red points indicate the position of the SNP identified in the AMP. Blue points indicate the SNP identified in the BMP. Candidate R genes annotated by DRAGO3, CGs annotated with defense functions, and CGs identified in defense-related modules by Ribone et al. [58] are indicated at the bottom of the panel. The identification of the genes was obtained from the HanXRQ.v2.0 genome.

**Table 1 plants-14-01187-t001:** Genome-wide associations for Sunflower Verticillium Wilt (SVW) resistance under natural infection conditions.

Trait	Chromosome	SNP	Position (HanXRQ.v1)	Minor Allele Frequency	*p*-Value	LOD	Effect	Effect S.E.	Proportion of Phenotypic Variance
DI.Flw	10	10511	201,901,887	0.136	1.24 × 10^−5^	4.907	−0.905	0.214	**0.282**
DI.Flw	10	10561	209,211,855	0.129	4.19 × 10^−5^	4.378	−0.840	0.211	0.239
DI.Flw	10	9300	59,101,609	0.114	6.65 × 10^−5^	4.177	−0.863	0.223	**0.142**
DI.AUDPC	10	10511	201,901,887	0.136	8.46 × 10^−6^	5.073	−0.408	0.095	**0.307**
DI.AUDPC	10	10561	209,211,855	0.129	2.42 × 10^−5^	4.616	−0.381	0.093	0.263
DI.AUDPC	10	10513	201,944,131	0.144	3.49 × 10^−5^	4.457	−0.392	0.097	0.296
DI.AUDPC	9	8537	137,675,420	0.091	4.47 × 10^−5^	4.350	−0.325	0.082	**0.141**
DI.AUDPC	8	6824	19,474,689	0.102	4.49 × 10^−5^	4.348	−0.307	0.077	0.131
DI.AUDPC	8	7394	76,546,603	0.091	4.74 × 10^−5^	4.324	−0.343	0.086	**0.157**
DI.AUDPC	10	10484	199,022,192	0.076	5.85 × 10^−5^	4.233	−0.675	0.172	0.236
DI.AUDPC	10	10452	196,358,971	0.144	8.65 × 10^−5^	4.063	−0.364	0.095	0.264
DS.Gf	10	10511	201,901,887	0.136	1.09 × 10^−5^	4.964	−0.357	0.084	**0.270**
DS.Gf	10	9300	59,101,609	0.114	2.87 × 10^−5^	4.542	−0.373	0.092	**0.162**
bDS.Gf	8	6824	19,474,689	0.102	1.13 × 10^−5^	4.945	−0.883	0.208	**0.143**
bDS.Gf	10	10511	201,901,887	0.136	1.99 × 10^−5^	4.701	−1.039	0.253	**0.262**
PC1	10	10511	201,901,887	0.136	6.20 × 10^−5^	4.208	−3.105	0.800	**0.294**

DI.Flw: Disease Incidence at flowering; DI.AUDPC: Area under the Disease Progress Curve for Disease Incidence; DS.Gf: Disease Severity at grain filling; bDS.Gf: binomial Disease Severity at grain filling; PC1: First Principal Component from a multivariate analysis of Disease Descriptors. The highest Proportions of Phenotypic Variance for each region are indicated in bold.

**Table 2 plants-14-01187-t002:** Composite Interval Mapping (CIM) of QTLs for Sunflower Verticillium Wilt (SVW) resistance under natural infection conditions.

Trait	QTL Name	Chr	SNP Closest to Refined QTL Peak	Pos (cM)	LOD	*p*-Value	% Variance Add. Model	Estimated Additive Effects	S.E of Estimated Effects	Pos Closest SNP HanXRQ v1	CI Left Marker Position (−1 LOD Drop)	CI Right Marker Position (−1 LOD Drop)	Interval (cM)
**DI.Flw**	**qSVW-10.1**	**10**	**86706_53**	**59**	**7.8**	**0,00**	**20.22%**	**−0.374**	**0.054**	**207,502,682**	**57.75**	**76.72**	**18.98**
**DI.Flw**	**qSVW-17.1**	**17**	**159418_28**	**259**	**5.96**	**2.00 × 10^−4^**	**11.70%**	**−0.296**	**0.056**	**171,251,880**	**255.23**	**269.18**	**13.94**
**DI.Flw**	**qSVW-14.1**	**14**	**129783_18**	**119**	**4.77**	**2.80 × 10^−3^**	**12.89%**	**−0.300**	**0.054**	**47,885,041**	**115.36**	**123.01**	**7.65**
DI.Flw	qSVW-08.1	8	63838_12	238	4.04	1.14 × 10^−2^	9.44%	−0.257	0.055	127,207,823	117.84	270.69	152.84
**DI.AUDPC**	**qSVW-10.1**	**10**	**86780_47**	**58**	**7.74**	**0,00**	**20.28%**	**−0.229**	**0.034**	**208,801,335**	**52.00**	**76.72**	**24.72**
**DI.AUDPC**	**qSVW-17.1**	**17**	**159430_35**	**257**	**5.7**	**2.00 × 10^−4^**	**13.36%**	**−0.185**	**0.033**	**171,446,989**	**134.61**	**262.59**	**127.98**
**DI.AUDPC**	**qSVW-08.2**	**8**	**63166_57**	**264**	**4.7**	**3.60 × 10^−3^**	**10.82%**	**−0.203**	**0.041**	**115,257,798**	**251.68**	**270.69**	**19.01**
DI.AUDPC	qSVW-14.1	14	129783_18	119	3.49	3.64 × 10^−2^	8.78%	−0.149	0.033	47,885,041	115.36	123.01	7.65
**DS.Gf**	**qSVW-10.1**	**10**	**86780_47**	**57.74**	**5.62**	**4.00 × 10^−4^**	**15.24%**	**−0.164**	**0.034**	**208,801,335**	**45.00**	**78.73**	**33.73**
**DS.Gf**	**qSVW-17.1**	**17**	**159418_28**	**258**	**4.13**	**9.60 × 10^−3^**	**11.27%**	**−0.132**	**0.032**	**171,251,880**	**125.00**	**262.00**	**137.00**
**bDS.Gf**	**qSVW-10.1**	**10**	**86923_8**	**75.71**	**7.33**	**0.00**	**20.67%**	**−0.460**	**0.071**	**211,290,639**	**48.00**	**78.73**	**30.73**
**bDS.Gf**	**qSVW-17.1**	**17**	**159418_28**	**258.90**	**7**	**0.00**	**18.73%**	**−0.445**	**0.072**	**171,251,880**	**255.23**	**262.59**	**7.35**
bDS.Gf	qSVW-14.1	14	129783_18	119.82	3.78	2.04 × 10^−2^	8.40%	−0.295	0.071	47,885,041	60.47	127.43	66.96
**PC1**	**qSVW-10.1**	**10**	**86780_47**	**57.75**	**7.06**	**0.00**	**17.60%**	**−1.578**	**0.258**	**208,801,335**	**48.00**	**78.73**	**30.73**
**PC1**	**qSVW-17.1**	**17**	**159418_28**	**259**	**5.8**	**2.00 × 10^−4^**	**15.59%**	**−1.502**	**0.261**	**171,251,880**	**242.64**	**329.77**	**87.13**
PC1	qSVW-14.1	14	129783_18	119.83	3.84	1.72 × 10^−2^	7.47%	−1.034	0.259	47,885,041	115.36	123.01	7.65
PC1	qSVW-13.1	13	121588_42	53	3.36	4.70 × 10^−2^	6.33%	0.942	0.257	72,945,739	15.23	114.12	98.89

DI.Flw: Disease Incidence at flowering; DI.AUDPC: Area under the Disease Progress Curve for Disease Incidence; DS.Gf: Disease Severity at grain filling; bDS.Gf: binomial Disease Severity at grain filling; PC1: First Principal Component from a multivariate analysis of Disease Descriptors. Rows in bold indicate QTLs obtained at alpha 1%.

**Table 3 plants-14-01187-t003:** Number of Candidate Genes (CGs) within 400 Kb windows around QTL-associated SNPs.

Mapping Method	SNP	XRQ2.0_pos	Number of CG
**GWAS**	7394	HanXRQr2Chr08_60565151	12
	9300	HanXRQr2Chr08_154480883	2
	8537	HanXRQr2Chr09_124441742	18
	10452	HanXRQr2Chr10_155523039	17
	10484	HanXRQr2Chr10_151936129	18
	10511	HanXRQr2Chr10_154638964	14
	10513	HanXRQr2Chr10_154681208	13
	10561	HanXRQr2Chr10_159899673	12
	6824	HanXRQr2Chr11_89733249	3
**CIM**	63166_57	HanXRQChr08-97375604	7
	63838_12	HanXRQChr08-110174623	12
	86706_53	HanXRQChr10-158829261	23
	86780_47	HanXRQChr10-159456303	10
	86923_8	HanXRQChr10-162279227	23
	121588_42	HanXRQChr13-69318860	6
	129783_18	HanXRQChr14-48685578	5
	159430_35	HanXRQChr17-159580342	3
	159418_28	HanXRQChr17-160161466	10

GWAS: Genome-Wide Association Study, CIM: Composite Interval Mapping.

**Table 4 plants-14-01187-t004:** Plant disease resistance gene (R gene) motifs identified in Candidate Genes (CGs) around QTL-associated SNPs.

Mapping Method	Associated SNP	Trait	CG ID (HanXRQv2.0)	R Class	Domain	Start of Domain	End of Domain
GWAS							
	8537	DI.AUDPC	HanXRQr2_Chr09g0388961	RLP	TM	1	15
			HanXRQr2_Chr09g0388961	RLP	TM	124	138
			HanXRQr2_Chr09g0388961	RLP	LRR	36	49
	10484	DI.AUDPC	HanXRQr2_Chr10g0452681	CN	NBS	157	172
			HanXRQr2_Chr10g0452681	CN	CC	4	24
			HanXRQr2_Chr10g0452681	CN	TM	170	189
	10511 10513	All Traits DI.AUDPC	HanXRQr2_Chr10g0453571	KIN	Kinase	19	40
			HanXRQr2_Chr10g0453571	KIN	Kinase	50	113
			HanXRQr2_Chr10g0453571	KIN	Kinase	115	225
			HanXRQr2_Chr10g0453581	T	TM	74	87
			HanXRQr2_Chr10g0453581	T	TM	150	179
			HanXRQr2_Chr10g0453581	T	TIR	16	97
			HanXRQr2_Chr10g0453581	T	TIR	103	125
			HanXRQr2_Chr10g0453581	T	TIR	143	157
			HanXRQr2_Chr10g0453591	KIN	TM	13	25
			HanXRQr2_Chr10g0453591	KIN	TM	361	384
			HanXRQr2_Chr10g0453591	KIN	Kinase	246	308
			HanXRQr2_Chr10g0453591	KIN	Kinase	317	420
			HanXRQr2_Chr10g0453591	KIN	Kinase	446	491
			HanXRQr2_Chr10g0453601	CK	CC	1	21
			HanXRQr2_Chr10g0453601	CK	TM	26	37
			HanXRQr2_Chr10g0453601	CK	Kinase	20	42
			HanXRQr2_Chr10g0453601	CK	Kinase	51	113
			HanXRQr2_Chr10g0453601	CK	Kinase	116	133
	10452	DI.AUDPC	HanXRQr2_Chr10g0453851	RLK	TM	6	20
			HanXRQr2_Chr10g0453851	RLK	TM	234	253
			HanXRQr2_Chr10g0453851	RLK	Kinase	287	389
			HanXRQr2_Chr10g0453851	RLK	Kinase	390	499
			HanXRQr2_Chr10g0453851	RLK	Kinase	517	571
			HanXRQr2_Chr10g0453851	RLK	LRR	84	211
	10452	DI.AUDPC	HanXRQr2_Chr10g0453961	N	NBS	152	162
			HanXRQr2_Chr10g0453961	N	TM	12	22
			HanXRQr2_Chr10g0453961	N	TM	94	104
**CIM**							
	86706_53	DI.Flw	HanXRQr2_Chr10g0454871	KIN	Kinase	137	164
			HanXRQr2_Chr10g0455011	KIN	Kinase	1	42
			HanXRQr2_Chr10g0455011	KIN	Kinase	47	155
			HanXRQr2_Chr10g0455011	KIN	Kinase	177	225
	159418_28	DI.Flw, Ds.Gf, bDS.Gf, PC1	HanXRQr2_Chr17g0822011	KIN	TM	230	245
			HanXRQr2_Chr17g0822011	KIN	Kinase	91	136
			HanXRQr2_Chr17g0822011	KIN	Kinase	168	202

DI.Flw: Disease Incidence at flowering; DI.AUDPC: Area under the Disease Progress Curve for Disease Incidence; DS.Gf: Disease Severity at grain filling; bDS.Gf: binomial Disease Severity at grain filling; PC1: First Principal Component from a multivariate analysis of Disease Descriptors. GWAS: Genome-Wide Association Study, CIM: Composite Interval Mapping.

## Data Availability

The original contributions presented in this study are included in the article/Appendix A. Further inquiries can be directed to the corresponding authors.

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
