# Peer review of "Combining Linkage and Association Mapping Approaches to Study the Genetic Architecture of Verticillium Wilt Resistance in Sunflower"

_plants, 2025, doi:10.3390/plants14081187_

Round 1
Reviewer 1 Report
Comments and Suggestions for Authors
- The introduction provides a strong rationale for studying SVW resistance, but it could benefit from a clearer explanation of why integrating QTL mapping and GWAS provides a more robust approach.
- The discussion identifies several candidate genes within the significant QTL regions. Providing more details on their biological function and how they contribute to SVW resistance would strengthen the interpretation.
- The manuscript states that 16 SNPs were significantly associated with SVW resistance, but the specific significance level (e.g., Bonferroni correction, false discovery rate adjustment) should be clearly mentioned.
- When discussing GWAS SNPs and QTL overlaps, briefly explain whether the SNPs within the same chromosome region are in high LD, supporting a shared genetic basis for resistance.
- While chromosome 10 is identified as the major region for SVW resistance, the paper should discuss whether the phenotypic variance explained by this locus is stable across different environments or genetic backgrounds.
- The study uses minor allele frequency (MAF) > 0.05 and a 50% missing data threshold for SNP selection. A brief justification for these cutoffs would enhance methodological clarity.
- The description of field trials and phenotypic evaluations is well-structured, but it would be beneficial to clarify how environmental variation was controlled during data collection.
- Discussing potential confounding factors (e.g., population structure, environmental influences, genotype-by-environment interactions) would help readers assess the study’s robustness.
- Some sections, particularly the QTL and GWAS results, contain repeated explanations. Consider streamlining these descriptions.
- Line 22-23: The sentence "we identified genomic regions associated with SVW resistance by integrating biparental and association mapping" should briefly mention the key result (chromosome 10 as the major resistance locus).
- Line 31-33: The statement "Both approaches consistently detected a major QTL on chromosome 10, explaining up to 30% of phenotypic variation" could specify which disease traits (e.g., incidence, severity) were most affected.
- Line 49-51: The claim that SVW can reduce yields by up to 30-73% should be supported with additional references to recent studies.
- Line 64-66: The prevalence of SVW in Argentina’s sunflower-growing regions should be linked to historical disease outbreaks or climate conditions favoring its spread.
- Line 81-84: The statement "Genetic resistance remains the most reliable approach to control this pathogen" should acknowledge that integrated disease management (e.g., cultural practices, chemical control) also plays a role.
- Line 131-135: The manuscript mentions five most informative disease descriptors used in mapping analysis. A short explanation of why these were chosen over others would be helpful.
- Line 257-262: When discussing the significant SNPs on chromosomes 8, 9, and 10, clarify whether these regions were previously linked to SVW resistance in past research.
- Line 339-343: The discussion of CHR08 QTLs could benefit from a comparison to other fungal resistance loci in sunflower to determine whether CHR08 is a general resistance region.
- Line 409-412: The identification of 356 candidate genes should highlight whether any are previously known defense-related genes in sunflower or other crops.
- Line 503-506: The conclusion that CHR10 is the dominant resistance locus could acknowledge whether this region is conserved across different sunflower populations or hybrids
Author Response
1. The introduction provides a strong rationale for studying SVW resistance, but it could benefit from a clearer explanation of why integrating QTL mapping and GWAS provides a more robust approach.
Response 1. Thank you for this suggestion. We recognize the importance of outlining the benefits of combining mapping strategies in the introduction. However, we believe these advantages are thoroughly addressed in the discussion (Lines 543–555 and 563–569), where we:
Highlight the improved mapping resolution of the region associated with disease control on Chr10 and the validation of results in different populations.
Present evidence for a comprehensive trait analysis that enhances the understanding of genetic architecture underlying complex traits.
Emphasize the importance of using different methods to capture the full spectrum of genetic variation—particularly noting that while single-locus GWAS did not identify a significant SNP on chromosome 17, other methods did (Lines 554–558).
2. The discussion identifies several candidate genes within the significant QTL regions. Providing more details on their biological function and how they contribute to SVW resistance would strengthen the interpretation.
Response 2. Thank you for your comment. The candidate genes (CGs) identified in this study exhibit diverse functions related to plant defense mechanisms. Specifically, these CGs play a central role in activating defense responses, including effector-triggered immunity, signal transduction, production of signaling molecules, and transmembrane transport. Collectively, these mechanisms have the potential to enhance sunflower resistance to SVW and offer a versatile approach to disease control.
Additionally, the functional annotation of the LACS2 gene (Lines 673–682) highlights its clear association with biological processes such as lateral root formation and suberin biosynthesis. Notably, suberin is a key component of the endodermis, which serves as a physical barrier against V. dahliae infection (Fradin et al., 2006; Fradin et al., 2009).
Ongoing transcriptomic studies from our group investigating the sunflower–V. dahliae interaction will further elucidate the role of CGs in defense processes, providing greater certainty regarding the genes involved in SVW resistance. We anticipate publishing these results soon.
Based on your suggestion, we have revised our conclusions to further emphasize the biological implications of our findings.
3. The manuscript states that 16 SNPs were significantly associated with SVW resistance, but the specific significance level (e.g., Bonferroni correction, false discovery rate adjustment) should be clearly mentioned.
Response 3. Thank you for this remark. The thresholds used are described in Methods in Lines 197-199, and in the first paragraph of the Results section.
4. When discussing GWAS SNPs and QTL overlaps, briefly explain whether the SNPs within the same chromosome region are in high LD, supporting a shared genetic basis for resistance.
Response 4. The LD patterns of the two MP under analysis, should differ significatively given their differences in genetic diversity (a diversity panel and a biparental RIL population). Nonetheless, several reports indicate the existence of large LD blocks in CHR10, being this the one with the slowest LD-decay among the 17 chromosomes (Mandel et al 2013 and Filippi et al. 2020). With this in consideration and the narrow regions explored by this study, it’s expected to have high LD among the SVW associated SNPs detected here.
5. While chromosome 10 is identified as the major region for SVW resistance, the paper should discuss whether the phenotypic variance explained by this locus is stable across different environments or genetic backgrounds.
Response 5. In terms of stability across different genetic backgrounds, the combination of these mapping approaches and the convergence of their results, especially in CHR10, confirms the stability of this region. In our previous publication (Montecchia et al. 2021), we investigated environmental variation by comparing different models for BLUES and BLUPs estimates across field trials.
6. The study uses minor allele frequency (MAF) > 0.05 and a 50% missing data threshold for SNP selection. A brief justification for these cutoffs would enhance methodological clarity.
Response 6. In this regard, we followed the same criteria as those implemented in Filippi et al. (2022). This clarification has now been incorporated into the text.
7. The description of field trials and phenotypic evaluations is well-structured, but it would be beneficial to clarify how environmental variation was controlled during data collection.
Response 7. Thank you for addressing this topic. We have not described these details in this manuscript as a thorough phenotypic analysis was covered in Montecchia et al. 2021 (lines 128 and 129).
8. Discussing potential confounding factors (e.g., population structure, environmental influences, genotype-by-environment interactions) would help readers assess the study’s robustness.
Response 8. In line with our previous comments, environmental influences and genotype-by-environment interactions were addressed in Montecchia et al. 2021. Regarding genetic structure, please refer to lines 184-197 in the Methods section. Please also refer to lines 256-259 in the Results section. In addition, this topic is also covered in lines 497-502 in the Discussion section.
9. Some sections, particularly the QTL and GWAS results, contain repeated explanations. Consider streamlining these descriptions.
Response 9. We agree that some paragraphs were redundant. The text has been streamlined as suggested.
10. Line 22-23: The sentence "we identified genomic regions associated with SVW resistance by integrating biparental and association mapping" should briefly mention the key result (chromosome 10 as the major resistance locus).
Response 10. Thank you for raising this point. The suggested changes have been incorporated, which required an adjustment to the text of the abstract to avoid repetition and keep the word count at 200.
11. Line 31-33: The statement "Both approaches consistently detected a major QTL on chromosome 10, explaining up to 30% of phenotypic variation" could specify which disease traits (e.g., incidence, severity) were most affected.
Response 11. Thank you for your comment. We believe that this sentence indeed conveys that PhV proportions of up to 30% were found primarily in DI-related traits and secondarily contributed to the reduction of DS. For clarity, we have emphasized this aspect in the manuscript at your suggestion.
12. Line 49-51: The claim that SVW can reduce yields by up to 30-73% should be supported with additional references to recent studies.
Response 12. We fully agree with your comment and it would be very helpful to have the more recent international publications on this subject. However, most of the literature on the distribution of inoculum in Argentina and the impact of the disease on yield dates from the early 2000s comes from the Argentine Sunflower Association (ASAGIR) and the National Institute of Agricultural Technology (INTA). Many of these publications are conference presentations by local phytopathologists or Spanish-language materials intended for farmers and agronomists. As noted in the quoted sentence, the reported yield reductions relate to highly susceptible material grown in heavily infested fields, not to current commercial hybrids. We have slightly revised this sentence to emphasize the episodic nature of these yield losses and added a reference in line 67 that summarizes several citations, including the INTA Plant Health Network (RETSAVE) reports that provide current prevalence data.
13. Line 64-66: The prevalence of SVW in Argentina’s sunflower-growing regions should be linked to historical disease outbreaks or climate conditions favoring its spread.
Response 13. We appreciate this suggestion and agree that historical disease outbreaks and climatic conditions have likely influenced the prevalence of SVW in Argentine sunflower-growing regions. As discussed in the manuscript, sunflower production has shifted over the last 60 years, particularly following the expansion of glyphosate-tolerant soybean, which displaced sunflower production to more marginal regions (Castaño 2018 [17]; Montecchia et al. 2021 [25]). This shift led to a concentration of sunflower cultivation in the southern province of Buenos Aires and eastern La Pampa, areas that overlap with Argentina’s primary potato-growing zone, where V. dahliae has an alternative host.
Given the pathogen’s mode of dissemination and the long-term persistence of its microsclerotia in the soil, crop rotations common in this region may have played a role in the emergence of the current distribution of SVW. However, as no formal analysis of these interactions is currently available, we prefer not to speculate further on their possible impact.
14. Line 81-84: The statement "Genetic resistance remains the most reliable approach to control this pathogen" should acknowledge that integrated disease management (e.g., cultural practices, chemical control) also plays a role.
Response 14. Thank you for this observation. In our previous report (Montecchia et al. 2021), we discussed this topic in more detail, highlighting the impact of no tillage on reducing the incidence of SVW in sunflower. In addition, crop rotation with non-host species has been proposed as a long-term strategy to reduce inoculum concentration in the soil (Castaño 2018; Quiroz et al. 2014).
Chemical control is ineffective against this disease and is not considered a management tool for this pathosystem in Argentina. Currently, genetic resistance remains the main criterion for the selection of hybrids by farmers. We have revised the sentence to more explicitly acknowledge additional aspects of disease control.
15. Line 131-135: The manuscript mentions five most informative disease descriptors used in mapping analysis. A short explanation of why these were chosen over others would be helpful.
Response 15. Thank you for this comment. We followed the criterion described in Montecchia et al. (2021) based on the potential impact of the disease on yield components at different phenological stages. Moreover, we included PC1 as a synthetic variable because it explains a high proportion of the multivariate variability captured by the 18 DDs evaluated in the study. Your suggestion has been included in lines 138–139.
16. Line 257-262: When discussing the significant SNPs on chromosomes 8, 9, and 10, clarify whether these regions were previously linked to SVW resistance in past research.
Response 16. In this section, we present our initial findings from the GWAS study and intentionally avoid referencing previous results. Instead, we discuss these results in the Discussion section, where we compare our results with the few existing molecular studies on this disease (lines 514–539).
17. Line 339-343: The discussion of CHR08 QTLs could benefit from a comparison to other fungal resistance loci in sunflower to determine whether CHR08 is a general resistance region.
Response 17. Thank you for your question. Given the novelty of these results and the recent report of another GWAS targeting SVW, we have decided to focus our discussion on these discoveries. As described in lines 534-539, the identified regions on CHR08 do not overlap with, nor are they close to, those found in previous reports. Nonetheless, we acknowledge the presence of several QTLs for fungal resistance on CHR8 (Guidini et al. 2022, Filippi et al. 2022, Ma et al. 2020, Fusari et al. 2012). However, the large distances between these QTLs and the QTL identified here preclude a direct comparison.
We also recognize that there are resistance loci for other fungi in the different chromosomes that host resistance to V. dahliae. However, the analysis of the responses of the PMA genotypes that host the different resistance regions has not provided evidence of multiple resistance loci in sunflower chromosomes. Exploring this aspect is very interesting, as is the study of the distances between loci. For both approaches, we need to deepen our investigations, and particularly for determining the distances, more genomic references of sunflower are required, as well as intensifying this knowledge in Argentine germplasm.
18. Line 409-412: The identification of 356 candidate genes should highlight whether any are previously known defense-related genes in sunflower or other crops.
Response 18. Considering the proposed method and the large number of candidate genes in this region of chromosome 10, we applied objective approaches to refine this group before comparing our results with previous findings. The procedure described in lines 468–482 allowed us to identify putative R genes and candidate genes (CGs) associated with defense response functions or processes, reducing the original set of 356 CGs to 39 CGs. These CGs have annotations or expression patterns associated with disease resistance in sunflower and other crops. The annotations for all candidate genes identified in this study can be found in Supplementary Table 10.
19. Line 503-506: The conclusion that CHR10 is the dominant resistance locus could acknowledge whether this region is conserved across different sunflower populations or hybrids.
Response 19. We have not confirmed that the major QTL region on CHR10 is conserved across different sunflower populations or hybrids, as this is beyond the scope of this paper. We hypothesize that CHR10 is a conserved region within the germplasm given the exhaustive efforts of breeding programs worldwide to find resistance to fungal diseases. This conclusion is also supported by the fact that the discovery of this region on chromosome 10 in the AMP of INTA with the observed significance levels confirms that it affects a very diverse material covering a large genetic diversity, similar to the diversity in other sunflower panels such as INRA or USDA-UBC (Filippi et al. 2020). Moreover, this region has been successfully mapped with genomic sunflower references such as HanXRQ.v2 and HA412-HO.v2, confirming its presence beyond any structural variation. However, testing this population with other V. dahliae strains or phytopathological races, on a race-specific approach, might alter the predominance of this region. In this study, we are not able to analyze the impact of V. dahliae races because the inoculum reservoirs studied were enriched over years with isolates from the sunflower growing region in Argentina.
Thank you very much for taking the time to review our manuscript and for your valuable comments. We greatly appreciate your insights and suggestions.
Reviewer 2 Report
Comments and Suggestions for Authors
The manuscript titled “Combining Linkage and Association Mapping Approaches to 2 Study the Genetic Architecture of Verticillium Wilt Resistance in Sunflower” is devoted to identification of genomic regions associated with resistance to sunflower Verticillium wilt and leaf mottle. It was done by integrating biparental and association mapping in populations of sunflowers from the Argentine National Institute of Agricultural Technology. Nine replicated field trials were conducted to assess disease incidence and severity. The reservoirs from highly infested V. dahliae Argentina’s sunflower-growing region were used to assess disease incidence and severity.
Both mapping populations were genotyped using double-digest restriction-site-associated DNA sequencing (ddRADseq), and association mapping was performed using 18,161 single-nucleotide polymorphisms (SNPs) with Single-Locus Mixed Linear Models (MLMs). Biparental quantitative trait locus (QTL) mapping based on a genetic map with 1,769 SNPs also identified key resistance loci.
Both approaches consistently detected a major QTL on chromosome 10, which accounted for up to 30% of the phenotypic variation in traits related to disease incidence and contributed to reducing disease severity.. In addition, QTLs on chromosomes 17, 8, 9, 14, 13, and 11 were also associated with reduced disease incidence and severity.
The manuscript reports accurate work done with modern genetic and genomic methods. Results are reliable and highly important for sunflower breeding.
There is only one question on Table 1. (Genome-wide associations for Sunflower Verticillium Wilt (SVW) resistance under natural infection conditions) content - Lines bDS.Gf and PC1 have identical figures in the first 4 columns, although other columns are different:
bDS.Gf 10 10511 201,901,887 0.136
PC1 10 10511 201,901,887 0.136
Is it correct data or mistyping?
Author Response
The manuscript titled “Combining Linkage and Association Mapping Approaches to 2 Study the Genetic Architecture of Verticillium Wilt Resistance in Sunflower” is devoted to identification of genomic regions associated with resistance to sunflower Verticillium wilt and leaf mottle. It was done by integrating biparental and association mapping in populations of sunflowers from the Argentine National Institute of Agricultural Technology. Nine replicated field trials were conducted to assess disease incidence and severity. The reservoirs from highly infested V. dahliae Argentina’s sunflower-growing region were used to assess disease incidence and severity.
Both mapping populations were genotyped using double-digest restriction-site-associated DNA sequencing (ddRADseq), and association mapping was performed using 18,161 single-nucleotide polymorphisms (SNPs) with Single-Locus Mixed Linear Models (MLMs). Biparental quantitative trait locus (QTL) mapping based on a genetic map with 1,769 SNPs also identified key resistance loci.
Both approaches consistently detected a major QTL on chromosome 10, which accounted for up to 30% of the phenotypic variation in traits related to disease incidence and contributed to reducing disease severity.. In addition, QTLs on chromosomes 17, 8, 9, 14, 13, and 11 were also associated with reduced disease incidence and severity.
The manuscript reports accurate work done with modern genetic and genomic methods. Results are reliable and highly important for sunflower breeding.
There is only one question on Table 1. (Genome-wide associations for Sunflower Verticillium Wilt (SVW) resistance under natural infection conditions) content - Lines bDS.Gf and PC1 have identical figures in the first 4 columns, although other columns are different:
bDS.Gf 10 10511 201,901,887 0.136
PC1 10 10511 201,901,887 0.136
Is it correct data or mistyping?
Response:
Thank you very much for taking the time to review our manuscript and for your valuable comments on our work. We are pleased to know that it has met your expectations.
To address your comment about Table 1, we can confirm that this is not an error. This information corresponds to (in order of appearance): Analyzed Trait, Chromosome, SNP ID, SNP position on HanXRQ.v1, and Minor Allele Frequency. This shows that the same SNP 10511, is associated with both binomial disease severity at grain filling (bDS.Gf) and principal component 1 (PC1). The following columns are different because the p-value at which the association was established, its LOD score, their additive effects, and the proportion of explained variance between variables varied for each independent association analysis.